# SOAP: Improving and Stabilizing Shampoo using Adam for Language Modeling

**Nikhil Vyas**[*]
Harvard University

**Depen Morwani**[*]
Harvard University

**Rosie Zhao**[†]
Harvard University

**Itai Shapira**[†]
Harvard University

**David Brandfonbrener**
Kempner Institute at Harvard University

**Lucas Janson**
Harvard University

**Sham Kakade**
Kempner Institute at Harvard University

## ABSTRACT

There is growing evidence of the effectiveness of Shampoo, a higher-order pre-conditioning method, over Adam in deep learning optimization tasks. However, Shampoo's drawbacks include additional hyperparameters and computational overhead when compared to Adam, which only updates running averages of first- and second-moment quantities. This work establishes a formal connection between Shampoo (implemented with the 1/2 power) and Adafactor — a memory-efficient approximation of Adam — showing that Shampoo is equivalent to running Adafactor in the eigenbasis of Shampoo's preconditioner. This insight leads to the design of a simpler and computationally efficient algorithm: **S**hampo**O** with **A**dam in the **P**reconditioner's eigenbasis (SOAP). With regards to improving Shampoo's computational efficiency, the most straightforward approach would be to simply compute Shampoo's eigendecomposition less frequently. Unfortunately, as our empirical results show, this leads to performance degradation that worsens with this frequency. SOAP mitigates this degradation by continually updating the running average of the second moment, just as Adam does, but in the current (slowly changing) coordinate basis. Furthermore, since SOAP is equivalent to running Adam in a rotated space, it introduces only one additional hyperparameter (the preconditioning frequency) compared to Adam. We evaluate SOAP on language model pre-training, with experiments on 360m and 660m sized models. In the large batch regime, SOAP reduces the number of iterations by over 40% and wall clock time by over 35% compared to AdamW, with approximately 20% improvements in both metrics compared to Shampoo. An implementation of SOAP is available at `https://github.com/nikhilvyas/SOAP`.

## 1 INTRODUCTION

With ever-increasing costs of LLM training, optimization efficiency has become a central question in the field of deep learning. Several recent works have tackled this challenge by addressing both the memory (Zhao et al., 2024a; Wang et al., 2024) and compute (Anil et al., 2020) footprint of optimizers. In Algoperf (Dahl et al., 2023), a recent optimization efficiency benchmark, Shampoo (Gupta et al., 2018a), a second-order algorithm, outperformed all other submissions, including Adam (Kingma & Ba, 2015), reducing wall-clock time by 28% (MLCommons, 2024). Higher-order preconditioning has also been applied in large-scale training runs, such as Gemini-1.5 Flash (Gemini Team, 2024).

The success of Shampoo has drawn increasing attention from the deep learning community. Several works have explored ways to scale Shampoo by improving its memory and compute efficiency

---

[*]Equal contribution. Correspondence to `nikhil@g.harvard.edu`.
[†]Equal contribution.

(Wang et al., 2024; Anil et al., 2020; Shi et al., 2023). Other research (Morwani et al., 2024) has examined the theoretical foundations of Shampoo and proposed minor adjustments (such as using power $1/2$ rather than $1/4$) that align with prior empirical findings (Anil et al., 2020). Moreover, Morwani et al. (2024) also showed that Shampoo with the aforementioned modifications is close to the optimal Kronecker approximation of the Adagrad (Duchi et al., 2011b) optimizer.

Our first contribution is demonstrating that the variant of Shampoo proposed by Morwani et al. (2024) is equivalent[1] to running Adafactor (Shazeer & Stern, 2018; Zhai et al., 2022) in the eigenbasis provided by Shampoo's preconditioner. This interpretation of Shampoo connects it to a broader family of methods (e.g. (George et al., 2018)) that design second-order algorithms by running a first-order method in the eigenbasis provided by a second-order method. Building on this insight, we can explore a broader design space for combining first and second order methods. Many of our design choices are a synthesis of conceptual ideas from prior works of George et al. (2018); Anil et al. (2020); Morwani et al. (2024) as well as implementation ideas from works of Wang et al. (2024); Zhao et al. (2024a).

Explicitly, we study SOAP (**S**hamp**O**O with **A**dam in the **P**reconditioner's eigenbasis) an algorithm that runs AdamW in the eigenbasis provided by Shampoo. Our main contributions are as follows:

- We make a formal connection between the Shampoo and the Adafactor algorithm. This insight leads us to consider the SOAP algorithm, which runs AdamW in the preconditioned space provided by Shampoo.
- SOAP outperforms both Shampoo and Adam in language model pre-training tasks with model sizes 360m and 660m, even after extensive hyperparameter tuning of Shampoo.
- SOAP reduces the number of hyperparameters compared to Shampoo, resulting in only one additional hyperparameter compared to AdamW: preconditioning frequency.
- SOAP demonstrates greater robustness to large preconditioning frequency compared to Shampoo on language model pre-training tasks.

We should also note that while similar algorithmic variants have been discussed in the literature (e.g. see the appendix of Anil et al. (2020)), we are the first to systematically evaluate it.

**Organization:** In Section 3, we discuss related works. In Section 4, we start by showing an equivalence between Shampoo (with exponent 1/2) and running Adafactor in the eigenspace given by Shampoo, then with this equivalence as the starting point we describe SOAP. In Section 5, we provide our experimental methodology and in Section 6, we compare the performance of AdamW, Shampoo and SOAP on language modeling tasks. In Appendices B.2 and B.3 we discuss the the space and time complexity of SOAP and how it can be improved. In Appendix C we show that efficiency benefits of SOAP over AdamW are maintained for longer duration runs where #tokens = $100 \times$ model size.

## 2 NOTATION AND BACKGROUND

We denote the weight matrix of a neural network layer by $W \in \mathbb{R}^{m \times n}$, and the corresponding gradient by $G \in \mathbb{R}^{m \times n}$. At a given time step $t$, these are denoted as $W_t$ and $G_t$, respectively. For a batch of inputs at time $t$, denoted by $B_t$, the loss and its gradient evaluated at $W_t$ are represented as $\phi_{B_t}(W_t)$ and $\nabla_W \phi_{B_t}(W_t)$, respectively.

Adagrad (Duchi et al., 2011b) is an online learning second-order algorithm that maintains a preconditioner $H \in \mathbb{R}^{mn \times mn}$. If the vectorized gradient at time $t$ is denoted by $g_t$ (i.e., $g_t = \text{vec}(G_t) \in \mathbb{R}^{mn}$), then the update of the preconditioner and the vectorized weights $w_t \in \mathbb{R}^{mn}$ with learning rate $\eta$ is given by

$$H_t = H_{t-1} + g_t g_t^\top; \quad w_t = w_{t-1} - \eta H_t^{-1/2} g_t$$

Adam (Kingma & Ba, 2015), a widely used first-order optimization algorithm in deep learning is a diagonal approximation of Adagrad. It maintains an exponential moving average of the gradients

---

[1]Given this connection, the results of Morwani et al. (2024) can be interpreted as showing that the eigenbasis provided by Shampoo's preconditioner is close to the "optimal" basis for running Adafactor.

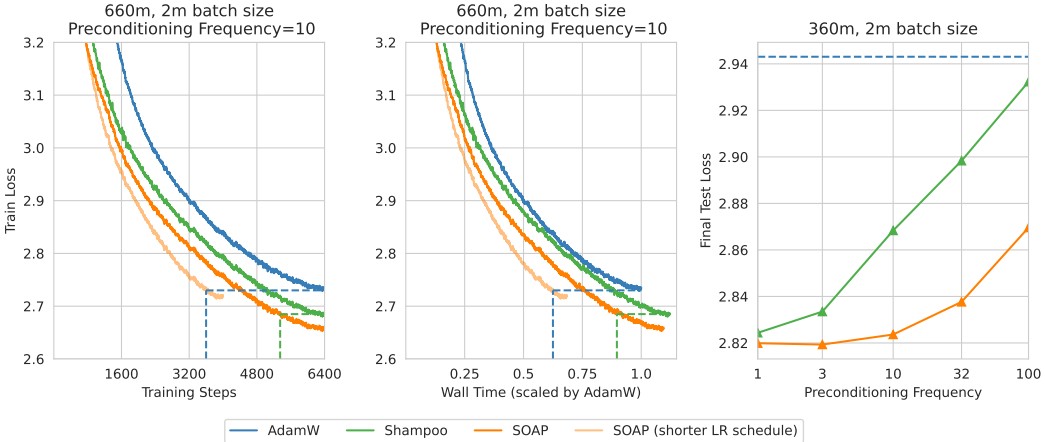

Figure 1: Comparing performance of tuned runs for AdamW, Shampoo (using DistributedSham-poo (Shi et al., 2023) implementation) and SOAP. In left and middle figures, Shampoo and SOAP use a preconditioning frequency of 10. The "shorter LR schedule" plot is where we tuned the cosine decay so as to achieve the same terminal performance as AdamW. There we observe a $\geq 40\%$ reduction in the number of iterations and a $\geq 35\%$ reduction in wall clock time compared to AdamW, and approximately a 20% reduction in both metrics compared to Shampoo. In the right figure we ablate preconditioning frequency and observe a slower degradation of performance of SOAP as compared to Shampoo. See Section 6 for a discussion of experimental results and ablation of batch size and Section 5 for experimental methodology.

$G_t$ (denoted as $M_t$) and of element-wise squared gradients $G_t^2$ (denoted as $V_t$) for a given weight matrix $W$. Its update rule with learning rate $\eta$ is given by

$$W_t \leftarrow W_{t-1} - \eta \frac{M_t}{\sqrt{V_t}},$$

where the division is performed element-wise.

Adafactor (Shazeer & Stern, 2018; Zhai et al., 2022), a variant of Adam, replaces $V_t$ with its best rank-1 approximation $V_t'$ to reduce memory usage. While the original Adafactor paper (Shazeer & Stern, 2018) proposed additional modifications, such as changes to the learning rate schedule, we focus on the version of Adafactor proposed in recent works (Zhai et al., 2022; Zhao et al., 2024c), whose update with learning rate $\eta$ is given by

$$W_t \leftarrow W_{t-1} - \eta \frac{M_t}{\sqrt{V_t'}}.$$

Shampoo (Gupta et al., 2018b) is a second-order optimization algorithm that approximates Adagrad and maintains two preconditioners, $L_t \in \mathbb{R}^{m \times m}$ and $R_t \in \mathbb{R}^{n \times n}$, for a given weight matrix $W \in \mathbb{R}^{m \times n}$. The updates for the preconditioners and the weights with learning rate $\eta$ are as follows:

$$L_t \leftarrow L_{t-1} + G_t G_t^T; \quad R_t \leftarrow R_{t-1} + G_t^T G_t; \quad W_t \leftarrow W_{t-1} - \eta L_t^{-1/4} G_t R_t^{-1/4}.$$

In practice, Shampoo is implemented with several other modifications such as layerwise learning rate grafting and exponents other than $-1/4$. We will use the DistributedShampoo (Shi et al., 2023) implementation which has these variations available as hyperparameters.

## 3 RELATED WORK

We begin by discussing works that are closely related, including George et al. (2018); Anil et al. (2020) and Zhao et al. (2024a). Subsequently, we cover extended related works.

**KFAC** (Martens & Grosse, 2015) is a well-known second-order optimization algorithm designed for neural networks. E-KFAC (George et al., 2018) builds upon KFAC in a manner analogous to our

extension of Shampoo, introducing a diagonal preconditioner that is updated between KFAC inversion steps. However, E-KFAC's algorithm is not identical to running Adam in KFAC's eigenbasis, as the diagonal preconditioner is not Adam.

Anil et al. (2020) introduced several algorithmic and numerical improvements to develop a practical and scalable version of Shampoo (Gupta et al., 2018b). Notably, they empirically found that using an exponent of $1/2$ outperforms the original exponent of $1/4$ in Shampoo. Of particular interest to our work is Appendix B of Anil et al. (2020), where, inspired by E-KFAC, they describe an algorithm that is essentially equivalent to SOAP for 2D layers. However, no experiments were provided, and the authors claimed that unpublished experiments showed no empirical improvement over Shampoo. This discrepancy between our findings may be due to some of the implementation details of SOAP.

**GaLore** (Zhao et al., 2024a) was recently proposed as a method to reduce Adam's memory footprint by maintaining momentum in a low-rank subspace derived from the singular value decomposition (SVD) of the gradients. Their algorithm's full-rank version bears similarity to ours, with some notable distinctions. Firstly, their projection subspace is determined by the SVD of the current gradient, while we maintain an exponential moving average of $GG^T$ and $G^TG$. Secondly, we retain momentum in the original space and project it onto the preconditioned space, whereas they maintain it in the preconditioned space and do not rotate it each time the preconditioned space is updated. In Appendix D, we study GaLore's performance and find that our modifications are necessary for improving upon Shampoo. Moreover, their method only projects one side of a layer using the eigenbasis while using the identity basis on the other side. We examine the impact of one-sided projection for SOAP in Appendix B.1.

**Diagonal Preconditioning based Optimizers:** Other than AdamW, there are other optimizers which involve diagonal preconditoning such as Lion (Chen et al., 2023), Sophia (Liu et al., 2024), and Adafactor (Shazeer & Stern, 2018). Recent works of Kaddour et al. (2023); Zhao et al. (2024c) showed that these optimizers perform comparably to AdamW for LLM pretraining but do not surpass it. This suggests the need to explore non-diagonal preconditioners. We discuss prior works on non-diagonal preconditioners below.

**Second-Order Optimization:** Research on second-order optimization in deep learning is generally divided into two categories: Hessian-free methods and methods that estimate the Hessian.

**Hessian-Free Methods:** Hessian-free approaches (Martens, 2010; Martens & Grosse, 2015) optimize without explicitly computing the Hessian matrix, instead employing iterative techniques to approximate the Newton step. Other recent works (Li, 2018; 2024; Pooladzandi & Li, 2024) have focused on designing iterative preconditioners to improve the convergence specifically for stochastic optimization algorithms.

**Hessian Estimation Methods:** These methods maintain an efficient approximation of the Hessian for neural networks. KFAC (Martens & Grosse, 2015) and Shampoo (Gupta et al., 2018b) are two widely recognized methods in this area.

KFAC (Martens & Grosse, 2015) was one of the first approaches to go beyond diagonal preconditioners in neural networks, demonstrating that a layer-wise Kronecker-factored preconditioner approximates the layer-wise Hessian in multi-layer perceptrons (MLPs). Subsequent works (Martens et al., 2018; Osawa et al., 2019) extended KFAC to other architectures. Recent research (George et al., 2018; Gao et al., 2021) has further improved trace and diagonal estimates for KFAC. Efforts to scale up KFAC (Ba et al., 2017; Puiu, 2022; 2023; Eschenhagen et al., 2023) have focused on making the inversion step more efficient or enhancing distributed implementations.

Shampoo (Gupta et al., 2018b), another second-order optimization algorithm, is motivated by the online learning algorithm Adagrad (Duchi et al., 2011a). Shampoo also employs a layer-wise Kronecker-factored preconditioner. A recent distributed implementation of Shampoo (Shi et al., 2023) won an optimization efficiency benchmark (Dahl et al., 2023), highlighting the practical utility of second-order methods in deep learning. Few recent works (Duvvuri et al., 2024; Morwani et al., 2024) have provided theoretical advancements on top of Shampoo. Other works (Anil et al., 2020; Peirson et al., 2022; Lin et al., 2024; Wang et al., 2024) have proposed various strategies to improve Shampoo's scalability. We defer a comparison of SOAP with these methods to future work.

## 4 ALGORITHM

### 4.1 THEORY

We begin by describing an equivalence between Shampoo and running Adafactor in the eigenbasis of the Shampoo preconditioner. For simplicity we omit momentum but the equivalence also holds with momentum. For this equivalence we use Shampoo with the following modifications from the original Shampoo optimizer (Gupta et al., 2018b):

1. We use power $1/2$ instead of power $1/4$. This was already recommended in practical implementations (Anil et al., 2020; Shi et al., 2023) and a theoretical connection between optimal Kronecker approximation of Adagrad (Duchi et al., 2011b) preconditioner and Shampoo with power $1/2$ was established in Morwani et al. (2024).

2. We also use the scalar correction to per layer learning rates described in Ren & Goldfarb (2021); Morwani et al. (2024).

3. Instead of the running average of $L$ and $R$ across time steps, we use dataset averages.

With these changes in place (first occurrence of these changes is highlighted in red in the algorithm below) we formally define the two algorithms whose equivalence we show in Algorithms 1 and 2.

---

**Algorithm 1** Single step of idealized Shampoo with power $1/2$.

1: Sample batch $B_t$.
2: $G_t \in \mathbb{R}^{m \times n} \leftarrow -\nabla_W \phi_{B_t}(W_t)$
3: $L \leftarrow \mathbb{E}_B[G_B G_B^T]$ {Where the expectation is over a random batch $B$.}
4: $R \leftarrow \mathbb{E}_B[G_B^T G_B]$
5: $\hat{H} \leftarrow L \otimes R / \text{Trace}(L)$
6: $W_t \leftarrow W_{t-1} - \eta \hat{H}^{-1/2} G_t = W_{t-1} - \eta L^{-1/2} G_t R^{-1/2} / \text{Trace}(L)^{-1/2}$

---

**Algorithm 2** Single step of idealized Adafactor in Shampoo's eigenspace.

1: Sample batch $B_t$.
2: $G_t \in \mathbb{R}^{m \times n} \leftarrow -\nabla_W \phi_{B_t}(W_t)$
3: $L \leftarrow \mathbb{E}_B[G_B G_B^T]$
4: $R \leftarrow \mathbb{E}_B[G_B^T G_B]$
5: $Q_L \leftarrow \text{Eigenvectors}(L)$
6: $Q_R \leftarrow \text{Eigenvectors}(R)$
7: $G'_t \leftarrow Q_L^T G_t Q_R$
8: {Idealized version of code for Adafactor taking $G'_t$ to be the gradient}
9: $G'_{B_t} \leftarrow Q_L^T G_{B_t} Q_R$
10: $A = \mathbb{E}_B[G'_B \odot G'_B] \mathbf{1}_m$ where $G'_B = Q_L^T G_B Q_R$
11: $C = \mathbf{1}_n^\top \mathbb{E}_B[G'_B \odot G'_B]$
12: $\hat{V}_t = \frac{AC^T}{\mathbf{1}_n^\top A}$ {Elementwise division}
13: $G''_t \leftarrow \frac{G'_t}{\sqrt{\hat{V}_t} + \epsilon}$ {Elementwise division and square root}
14: $G'''_t \leftarrow Q_L^T G''_t Q_R$ {Projecting back to original space}
15: $W_t \leftarrow W_{t-1} - \eta G'''_t$

---

**Claim 1.** *Algorithms 1 and 2 are equivalent.*

*Proof.* Consider $G_t$ in the basis created after rotating by $Q_L, Q_R$ i.e. $G'_t = Q_L^T G_t Q_R$. Let the eigenvalues of $\mathbb{E}_{B_t}[G_{B_t} G_{B_t}^T]$ and $\mathbb{E}_{B_t}[G_{B_t}^T G_{B_t}]$ be given by $\lambda_1, ..., \lambda_m$ and $\mu_1, ..., \mu_n$ respectively. Algorithm 1 scales the $i, j$ coordinate by $(\lambda_i \mu_j / (\sum_i \lambda_i))^{-1/2}$, while Algorithm 2 scales them by $(A_i C_j / (\sum_i A_i))^{-1/2}$. We now show that $A_i = \lambda_i$, an analogous argument shows $C_j = \mu_j$.

---

**Algorithm 3** Single step of SOAP for a $m \times n$ layer. Per layer, we maintain four matrices: $L \in \mathbb{R}^{m \times m}, R \in \mathbb{R}^{n \times n}$ and $V, M \in \mathbb{R}^{m \times n}$. For simplicity we ignore the initialization and other boundary effects such as bias correction. Hyperparameters: Learning rate $\eta$, betas $= (\beta_1, \beta_2)$, epsilon $\epsilon$, and preconditioning frequency $f$.
An implementation of SOAP is available at `https://anonymous.4open.science/status/SOAP-F93B`.

---

1: Sample batch $B_t$.
2: $G \in \mathbb{R}^{m \times n} \leftarrow -\nabla_W \phi_{B_t}(W_t)$
3: $G' \leftarrow Q_L^T G Q_R$
4: $M \leftarrow \beta_1 M + (1 - \beta_1)G$
5: $M' \leftarrow Q_L^T M Q_R$
6: {Now we "run" Adam on $G'$}
7: $V \leftarrow \beta_2 V + (1 - \beta_2)(G' \odot G')$ {Elementwise multiplication}
8: $N' \leftarrow \frac{M'}{\sqrt{\hat{V}_t}+\epsilon}$ {Elementwise division and square root}
9: {Now that we have preconditioned by Adam in the rotated space, we go back to the original space.}
10: $N \leftarrow Q_L N' Q_R^T$
11: $W \leftarrow W - \eta N$
12: {End of gradient step, we now update $L$ and $R$ and possibly also $Q_L$ and $Q_R$. }
13: $L \leftarrow \beta_2 L + (1 - \beta_2)GG^T$
14: $R \leftarrow \beta_2 R + (1 - \beta_2)G^T G$
15: **if** t % f == 0 **then**
16:     $Q_L \leftarrow$ `Eigenvectors`$(L, Q_L)$
17:     $Q_R \leftarrow$ `Eigenvectors`$(R, Q_R)$
18: **end if**

---

$$
\begin{aligned}
A_i &= e_i^T \mathbb{E}_B[G'_B \odot G'_B]\mathbf{1}_m \\
&= \mathbb{E}_B[\sum_j (G'_B)^2_{i,j}] \\
&= \mathbb{E}_B[\sum_j (u_i^T(G_B)v_j)^2] && \text{(Using definition of } G') \\
&= \mathbb{E}_B[\|u_i^T(G_B)\|^2] && (v_j \text{ form a basis)} \\
&= \mathbb{E}_B[u_i^T G_B G_B^T u_i] \\
&= \lambda_i && \text{(By definition of } \lambda_i \text{ and } u_i)
\end{aligned}
$$

$\square$

While these two algorithms are equivalent in their idealized forms, practical considerations reveal some differences. Firstly, the algorithms differ when using running averages instead of dataset averages. Secondly, and more significantly in practice, we do not invert or compute the eigenvector decomposition of $L$ and $R$ at every step. This means that the "adaptivity" of learning rates in Shampoo is limited[2] to the updates of $L$ and $R$. In contrast, with Adafactor in Shampoo's eigenspace, the second moment estimates (i.e., $A$ and $C$ in Algorithm 2) can be updated at every step as they are computationally inexpensive. Additionally, instead of using Adafactor, we can opt[3] for Adam, which offers more generality. Combining these insights leads to Algorithm 3 which can be interpreted as running Adam in Shampoo's eigenspace.

---

[2]We note that practical implementations of Shampoo use grafting which allows for learning rate adaptivity at every step, but this adaptivity is restricted to a single scalar per layer.

[3]Though using AdamW over Adafactor only gives very small improvements in performance, see Figure 5 and Appendix B.2. We also note that one can use any other diagonal preconditioner based optimizer in place of Adam, such as Lion (Chen et al., 2023), Sophia (Liu et al., 2024) or Schedule-Free AdamW (Defazio et al., 2024).

---

**Algorithm 4** `Eigenvectors` function, implemented using power iteration and QR decomposition. Inputs: PSD matrix $P$ and estimate of eigenvectors $Q$. If the estimate was exact we would have $P = QDQ^T$ where $D$ is the diagonal matrix with eigenvalues.

---
1: $S \leftarrow PQ$
2: $Q \leftarrow \text{QR}(S)$

---

We now describe some additional implementation details:

1. Algorithm 3 describes the behavior of the algorithm for 2D layers. Following Zhao et al. (2024a), for 1D layers we run standard AdamW. This reduces the overhead as compared to standard implementations of Shampoo which solve an eigenvector problem for 1D layers too.

2. Following Wang et al. (2024), we compute eigenvectors of $L$ (and $R$) using one step of power method (Algorithm 4). This requires doing one matrix multiplication followed by QR decomposition. QR decomposition is faster (Documentation, 2024) than standard eigenvector decomposition in PyTorch. For the first iteration, eigenvectors are initialized by doing a standard eigenvector decomposition.

3. For layers with huge dimensions such as the first and last layer in language modeling transformers, maintaining the eigenvectors would be space and time prohibitive. For such dimensions we fix the rotation matrix ($Q_L$ or $Q_R$) to be identity. Note that if we fix both $Q_L$ and $Q_R$ to be identity for a 2D layer, we would recover Adam.

4. Algorithm 3 omits bias correction and weight decay for simplicity, but these are used in the actual implementation, identical to their use in AdamW.

The main focus of the next sections will be to explore the empirical performance of this algorithm and its variations. In Appendices B.2 and B.3 we discuss the the space and time complexity of SOAP and how it can be improved.

## 5 EXPERIMENTAL METHODOLOGY

**Hyperparameter tuning:** We begin with hyperparameter values suggested by prior research for both AdamW and Distributed Shampoo (e.g., $\beta_2 = 0.95$). Initially, we conduct a learning rate sweep to determine the optimal learning rate for each optimizer. Once the optimal learning rate is identified, we perform two-dimensional sweeps for each of the remaining hyperparameters, where we vary the selected hyperparameter alongside the learning rate. The purpose of these sweeps is to demonstrate that our default hyperparameter settings are near-optimal, disregarding potential interactions between two non-learning-rate hyperparameters. A detailed discussion of the hyperparameter sweeps is provided in Appendix A.

**Throughput Measurement:** We evaluate the throughput of each optimizer by measuring the number of tokens processed per second. At present, we perform these measurements on a single H100 GPU and utilize gradient accumulation to accommodate large batch sizes. While this approach may seem to disadvantage AdamW— as the overhead of Shampoo/SOAP is compared against multiple gradient accumulation steps— it is important to note that the overhead of Shampoo/SOAP can be amortized across layers by distributing the updates across multiple GPUs. This technique is employed in the distributed implementation of Shampoo (Shi et al., 2023). A comprehensive comparison of distributed implementations of these algorithms is left to future work.

**Efficiency Benefits:** Simply running SOAP for the same duration as Shampoo and AdamW cannot be directly used to calculate the efficiency benefit (in terms of training steps or wall-clock time) of using SOAP since we use a cosine schedule. Therefore, we run SOAP on $.5, .625, .75$ and $.875$ fraction of the training data and fit a scaling law of the form $a + bN^{-\beta}$ through the final losses obtained, where $N$ represents the number of training points and $a, b, \beta$ are the parameters of the fit. We show these points and the corresponding scaling laws obtained in Figure 2. This scaling law is then used to calculate the efficiency benefit in terms of training steps and wallclock time as shown in Figure 2. Here, the horizontal lines represent the final losses of AdamW and Shampoo.

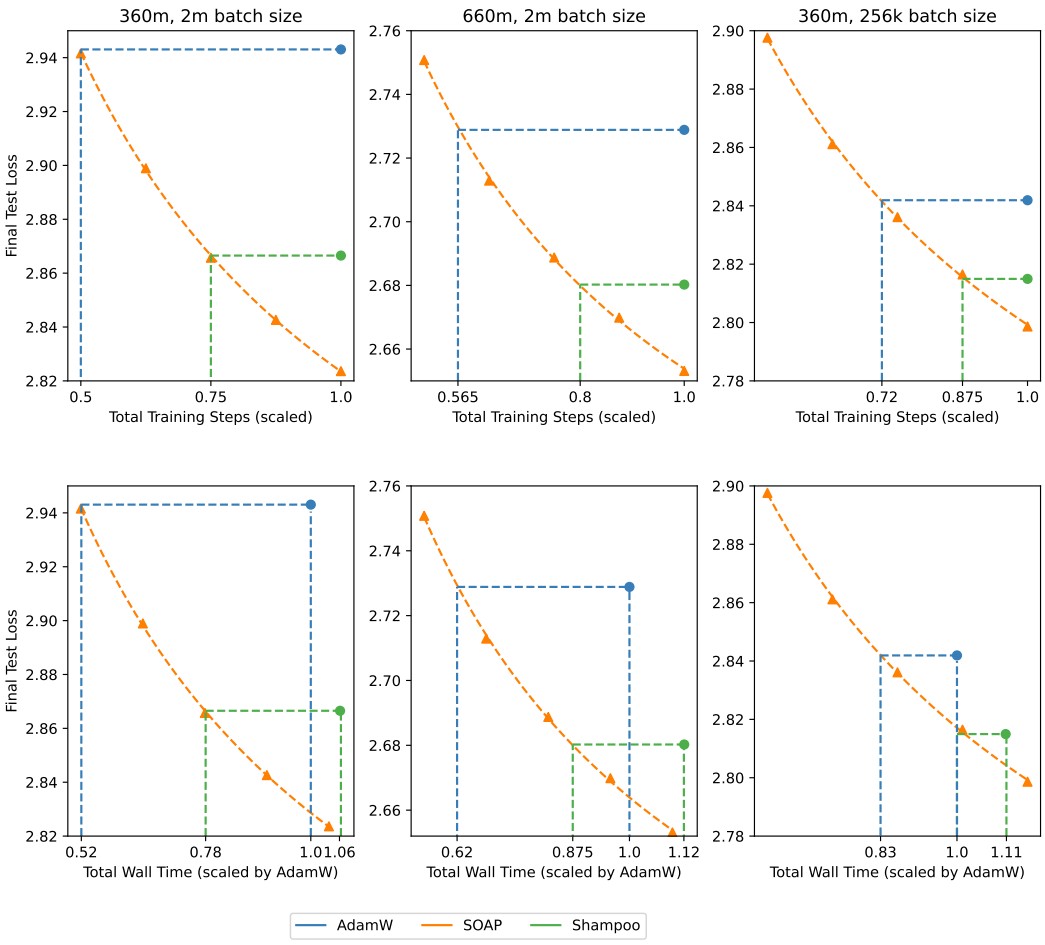

Figure 2: Precise efficiency benefits of SOAP over AdamW and Shampoo for 360m (at 256k and 2m batch size) and 660m (at 2m batch size) model. For the precise methodology, refer to Section 5.

# 6 LANGUAGE MODELING EXPERIMENTS

In this section we focus on empirically comparing AdamW, DistributedShampoo, and SOAP on language modeling tasks.

## 6.1 MEASURING EFFICIENCY BENEFITS

In Figure 1 (left and middle) and Figure 3 we show train loss curves for AdamW, Shampoo, and SOAP on 360m and 660m models with 2m token batch size and "chinchilla-optimal" i.e. 20x model size number of tokens. In these plots we observe that SOAP outperforms the other two optimizers. To directly calculate the efficiency benefit of SOAP, we also run SOAP with cosine decay for a shorter lr schedule, as shown in Figures 1 and 3. This allows us to approximate the following efficiency benefits (when batch size is set to 2m and preconditioning frequency to 10): $\geq 40\%$ reduction in the number of iterations and $\geq 35\%$ reduction in wall clock time compared to AdamW; $\approx 20\%$ reduction in iterations and wall clock time as compared to Shampoo. Precise efficiency benefit calculations are presented in Figure 2(left and middle). In Appendix C we show that efficiency benefits of SOAP over AdamW are maintained for longer duration runs where #tokens $= 100 \times$ model size.

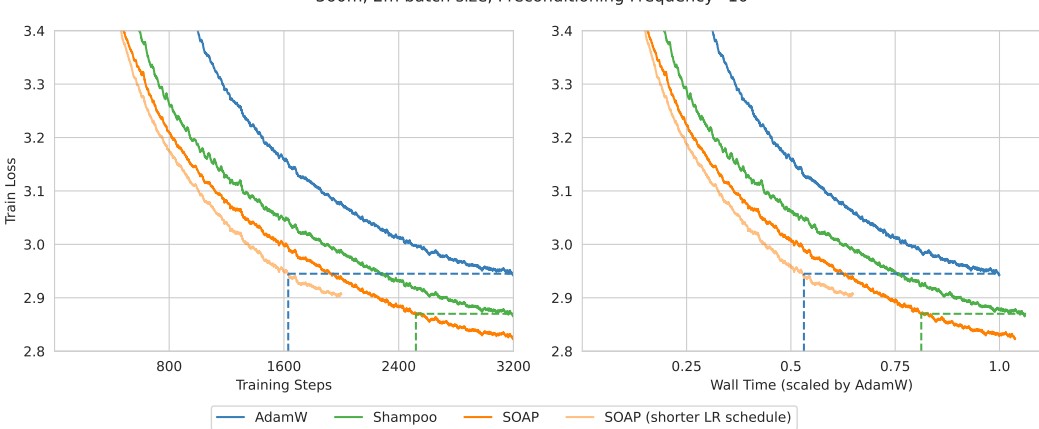

Figure 3: Comparing performance of tuned runs for AdamW, Shampoo (using DistributedShampoo (Shi et al., 2023) implementation) and SOAP. Shampoo and SOAP use preconditioning frequency of 10. We observe a $\geq 40\%$ reduction in the number of iterations and a $\geq 35\%$ reduction in wall clock time compared to AdamW, and approximately a $20\%$ reduction in both metrics compared to Shampoo. See Figure 1 for 660m results, Sections 6.2 and 6.3 for ablations of preconditioning frequency and batch size respectively, and Section 5 for detailed calculation of efficiency improvement and experimental methodology.

## 6.2 EFFECT OF FREQUENCY OF FINDING EIGENVECTORS/INVERSE

In Figure 1 (right), we compare SOAP and Shampoo with respect to preconditioning frequency. We observe the following:

- For all frequencies we tried from 1 to 100, both optimizers outperform AdamW.

- At frequency 1, SOAP and Shampoo are quite close in performance.

- At higher frequencies, the performance of both SOAP and Shampoo degrades but SOAP's performance degrades significantly slower than Shampoo's.

## 6.3 SOAP IMPROVES THE CRITICAL BATCH SIZE

When scaling up batch sizes, the ideal outcome is that doubling the batch size results in halving the number of training steps needed to achieve the same performance. The batch size at which this ideal scaling starts to break down is referred to by McCandlish et al. (2018) as the *critical batch size*. As models and datasets grow larger, it becomes increasingly important to develop optimizers that support larger critical batch sizes, thereby reducing the serial runtime of a training run. In this subsection, we compare the critical batch sizes of AdamW and SOAP. Relative to our baseline setup of a 2 million batch size, when we decrease the batch size by a factor of $k$, we increase the preconditioning frequency by the same factor. This ensures that the FLOPS and wall clock multiplicative overhead for the eigenvector decomposition steps remains consistent with the 2 million batch size setting.

We start by training a 360 million parameter model with a batch size of 256k for a "Chinchilla-optimal" number of tokens (20 times the model size) using AdamW, achieving a loss of 2.842. This value is set as the target loss for our comparisons. In Figure 4 (left), we show the number of steps AdamW and SOAP require to reach this target loss as we vary the batch size. SOAP consistently requires fewer steps across all batch sizes, with the multiplicative benefits becoming more pronounced at larger batch sizes. Additionally, we compare these results to the ideal scenario (dashed line) of linear scaling, where doubling the batch size halves the number of steps. SOAP more closely follows the linear scaling trend compared to AdamW, indicating that it has a higher critical batch size in this setup.

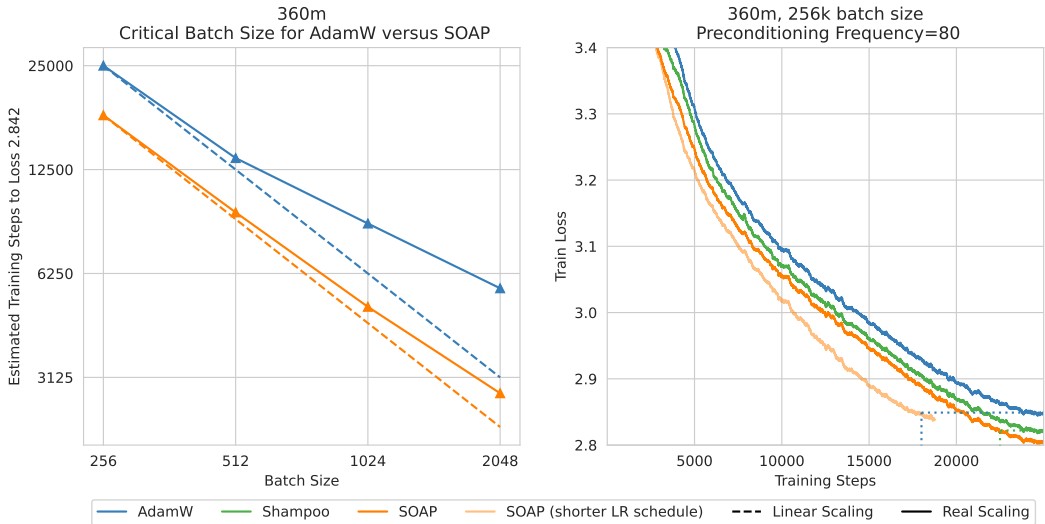

Figure 4: (left) Comparing the critical batch size of AdamW vs SOAP. We can see that SOAP improves the critical batch size, by being much closer to the ideal linear scaling with batch size as compared to AdamW. (right) Comparing performance of tuned runs for AdamW, Shampoo (using DistributedShampoo (Shi et al., 2023) implementation) and SOAP for token batch size of 256k. Shampoo and SOAP use preconditioning frequency of 80. We observe a $\geq 25\%$ reduction in the number of iterations compared to AdamW, and approximately a 10% reduction compared to Shampoo. See Figure 2 (right) for wall-clock time improvement and Section 5 for detailed calculation of efficiency improvement.

In Figure 4 (right), we present the optimal runs for each optimizer (including Shampoo) at the smallest batch size we consider: 256k. SOAP outperforms both Shampoo and AdamW, reducing the number of iterations by 25% compared to AdamW, and by approximately 10% compared to Shampoo. Furthermore, in Figure 2 (right, bottom), we demonstrate that SOAP also achieves a wall-clock time improvement of $\geq 15\%$ over AdamW and around 10% over Shampoo. We note that these results are a preliminary analysis for smaller batch size runs. Our approach of keeping the product of batch size and preconditioning frequency constant may not be optimal, and a better trade-off could likely be found. Furthermore, SOAP's overhead could potentially be reduced by performing $L$ and $R$ updates in lower precision (instead of fp32). Finally, the diminished efficiency gains of second-order methods at smaller batch sizes are consistent with prior findings (Zhang et al., 2019; Ishikawa & Yokota, 2024).

## 7  DISCUSSION AND LIMITATIONS

We study an optimizer called SOAP: **S**hampo**O** with **A**dam in the **P**reconditioner's eigenbasis. We show that SOAP outperforms both AdamW and Shampoo in language modeling tasks and show that it is more robust to changes in preconditioning frequency than Shampoo. While we have explored many factors such as batch size (Section 6.3) and training duration (Appendix C) we acknowledge that our study focuses on a relatively small scale compared to recent LLMs Touvron et al. (2023) which are two orders of magnitude bigger. We hypothesize that our findings on the performance of SOAP would generalize to larger scales due to its theoretical foundation. SOAP's robustness is also supported by the fact that SOAP is equivalent to running Adam in a rotated space, and Adam has proven to be effective across scale and tasks. However, this hypothesis remains to be validated.

For future work, we aim to improve the design of SOAP further, particularly by exploring the use of lower precision for preconditioners and optimizing its distributed implementation. Additionally, we are interested in testing SOAP's performance in other domains, such as vision, to evaluate its performance across different types of tasks.

## ACKNOWLEDGMENTS

SK, DM, and RZ acknowledges support from the Office of Naval Research under award N0001422-1-2377 and the National Science Foundation Grant under award #IIS 2229881. This work has been made possible in part by a gift from the Chan Zuckerberg Initiative Foundation to establish the Kempner Institute for the Study of Natural and Artificial Intelligence. NV, DM and RZ are supported by a Simons Investigator Fellowship, NSF grant DMS-2134157, DARPA grant W911NF2010021,and DOE grant DE-SC0022199. LJ acknowledges funding from the National Science Foundation DMS-2134157.

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

## A  EXPERIMENTAL SETUP

Many aspects of our setup such as models are the same as in Zhao et al. (2024c). We will restate those details verbatim for completeness.

We train language models on C4 tokenized with the T5 tokenizer (Raffel et al., 2020) and report results in terms of validation loss.

**Models.**    We start from the OLMo codebase (Groeneveld et al., 2024) and train decoder-only transformer models of three sizes: 210m, 360m, and 660m, where the parameter count refers to non-embedding parameters. The models have widths of 1024, 1024, and 1408 and depths of 12, 24, 24. We used the 210m model to explore various ablations, most of our reported results are on 360m and 660m. The MLP hidden dimension is 4x of the width. The activation function is GeLU (Hendrycks & Gimpel, 2016). We use RoPE positional encodings (Su et al., 2024). Attention heads are always dimension 64. We use PyTorch default LayerNorm. We use QK layer norm (Dehghani et al., 2023). Following Wortsman et al. (2024) we do not learn biases for the linear layers or LayerNorms. We train in mixed precision with bfloat16.

**Algorithms.**    We use the standard Pytorch implementation of AdamW (Paszke et al., 2019), the DistributedShampoo Shi et al. (2023) implementation of Shampoo. We implement ourselves SOAP and GaLore starting from an older version of Pytorch implementation of AdamW and the official GaLore implementation Zhao et al. (2024b).

**Default hyperparameters.**    We use $\beta_1 = 0.95$, as we found it to outperform $\beta_1 = 0.9$ in our sweeps for the 360m model. Following Wortsman et al. (2024) we use decoupled weight decay with coefficient $1e-4$ and z-loss with coefficient $1e-4$. We use the default value of $\epsilon = 1e-8$ in AdamW (actual or when used for grafting), SOAP and GaLore. We use warmup followed by cosine decay as our scheduler. We start the warmup and end the cosine decay at $0.1x$ the maximum learning rate.

**Default hyperparameters for DistributedShampoo**    Shi et al. (2023) state that they find the optimal exponent to be either $-1/2$ or $-1.82/4 \approx -1/2.2$. Our preliminary findings were similar to this. Hence we set the default values of exponent to be $-1/2.5$ for both 1D and 2D parameters. We set $\epsilon_{\text{shampoo}} = 1e-12$ and $\beta_{\text{shampoo}} = 0.95$ based on our initial set of experiments on the 210m model.

**Default hyperparameters for GaLore**    GaLore introduces an additional hyperparameter called scale ($\alpha$) since due to low rank updates the overall update magnitude decreases. Since we are running a full rank version of GaLore we set $\alpha = 1$.

**Token counts.**    For all of our runs we use a sequence length of 1024. For all models (except in Section 6.3), we use a token batch size of 2048k $\approx$ 2m. We default to training models for the approximately "chinchilla optimal" (Hoffmann et al., 2022) number of tokens that is $\approx$20 times the number of parameters. Explicitly, this means for our default batch size of 2m, the 210m models are trained for 1600 steps or $\approx$ 3.3b tokens. The 360m models are trained for 3200 steps, the 660m models are trained for 6400 steps.

### A.1  SWEEPING OVER HYPERPARAMETERS

**AdamW, 2m batch size:** Starting from the default hyperparameters above we do the following sweeps:

1. We sweep over learning rate in $\{.1, .0316, .01, \ldots, 3.16e-4\}$.

2. (360m) We sweep over the cross product of best 3 learning rates and $\beta_1 \in \{0.9, 0.95, 0.99\}$.

3. (360m) We sweep over the cross product of best 3 learning rates and $\beta_2 \in \{0.9, 0.95, 0.99\}$.

The last two of the sweeps did not yield any benefit for the 360m model with 2m batch size hence we only sweep over learning rate for the 660m model with 2m batch size.

**DistributedShampoo, 2m batch size:** Starting from the default hyperparameters above we do the following sweeps:

1. We sweep over learning rate in $\{.1, .0316, .01, \ldots, 3.16\mathrm{e}{-4}\}$.
2. (360m) We sweep over over the cross product of best 3 learning rates from above and $\epsilon_{\text{shampoo}} \in \{1\mathrm{e}{-11}, 1\mathrm{e}{-12}, 1\mathrm{e}{-13}\}$.
3. (360m) We sweep over over the cross product of best 3 learning rates from above and $\beta_{\text{shampoo}} \in \{.9, .95, .975\}$.
4. Let $e_1, e_2$ denote the exponents used in DistributedShampoo for 1D and 2D parameters respectively. We also sweep over the cross product of best 3 learning rates from above and $(e_1, e_2)$ in $\{(2, 2), (2.5, 2.5), (3, 3), (2, 4)\}$.

These sweeps did not yield any significant improvement in performance ($< .004$) for the 360m model. Hence we only sweep over the learning rate for the 660m model.

**SOAP, 2m batch size:** Starting from the default hyperparameters above we sweep over learning rate in $\{.1, .0316, .01, \ldots, 3.16\mathrm{e}{-4}\}$.

**AdamW, 256k batch size:** For the 360m model with 256 batch size we start from the default hyperparameters and do the following sweeps:

1. We sweep over learning rate in $\{.1, .0316, .01, \ldots, 3.16\mathrm{e}{-4}\}$.
2. We sweep over the cross product of best 3 learning rates and $\beta_2 \in \{0.95, 0.99\}$.

In the second sweep we observe small improvements in performance by using $\beta_2 = .99$, so our final numbers use $\beta_2 = .99$. This (small) improvement in performance by using a larger $\beta_2$ at smaller batch sizes was also observed by Porian et al. (2024); Zhao et al. (2024c).

**DistributedShampoo, 256k batch size:** For the 360m model with 256 batch size we start from the default hyperparameters and do the following sweeps:

1. We sweep over learning rate in $\{.1, .0316, .01, \ldots, 3.16\mathrm{e}{-4}\}$.
2. We sweep over the cross product of best 3 learning rates and $(\beta_2, \beta_{\text{shampoo}}) \in \{(.95, .95), (.99, .99)\}$.

In the second sweep we observe small improvements in performance by using $\beta_2 = \beta_{\text{shampoo}} = .99$, so our final numbers use $\beta_2 = \beta_{\text{shampoo}} = .99$.

**SOAP, 256k batch size:** For the 360m model with 256 batch size we start from the default hyperparameters and do the following sweeps:

1. We sweep over learning rate in $\{.1, .0316, .01, \ldots, 3.16\mathrm{e}{-4}\}$.
2. We sweep over the cross product of best 3 learning rates and $\beta_2 \in \{.95, .99\}$.

In the second sweep we observe small improvements in performance by using $\beta_2 = .99$, so our final numbers use $\beta_2 = .99$.

**Preconditioning frequency sweeps:** For the preconditioning frequency experiments of SOAP and Shampoo ( Figure 1 (right)), for each frequency we do a learning rate sweep over the best 3 learning rates found at preconditioning frequency 10. Other hyperparameters are set to their optimal values obtained using the precondition frequency 10 sweeps.

**360m and 660m shorter runs:** For each of the shorter runs of 360m and 660m models for the SOAP optimizer (Figure 2), we did learning rate sweep over the best 3 learning rates found for the standard length run. Other hyperparameters are set to their optimal values obtained using the standard length run.

**Warmup:** The warmup duration for the 360m and 660m models were 600 and 1200 steps respectively. For the shorter runs (Figure 2), for 360m model, the warmup durations were 400, 400, 500 and 525 steps for 0.5, 0.625, 0.75 and 0.875 runs respectively. For the 660m model, the warmup durations were 600, 750, 900 and 1050 steps for 0.5, 0.625, 0.75 and 0.875 runs respectively. For 360m model with 256k batch size (Section 6.3) we use a warmup for 4000 steps (total steps is 25000).

## B  FURTHER EFFICIENCY IMPROVEMENTS

In this section, we discuss space and time complexity of SOAP and provide an overview of potential avenues for further space and compute efficiency improvements in SOAP.

### B.1  ONE SIDED EIGENBASIS

As described in Section 3, Zhao et al. (2024a) have an algorithm similar to ours. One of the differences is that they only project the smaller side of the layer using the eigenbasis while using identity as the rotation matrix for the larger side i.e. if $m < n$ we set $Q_R = I_n$ in Algorithm 3 and if $m > n$ we set $Q_L = I_m$. Doing this leads to a reduction in space usage as well as reduction of optimizer time overhead, which is discussed in Appendices B.2.1 and B.3.1.

In Figure 5, it is evident that the one-sided projection results in slightly reduced performance compared to the original SOAP optimizer. However, it still performs on par with, or marginally better than, Shampoo, while maintaining greater computational efficiency. Further investigation into the potential for these variants to surpass the computational efficiency of original SOAP optimizer is left for future work.

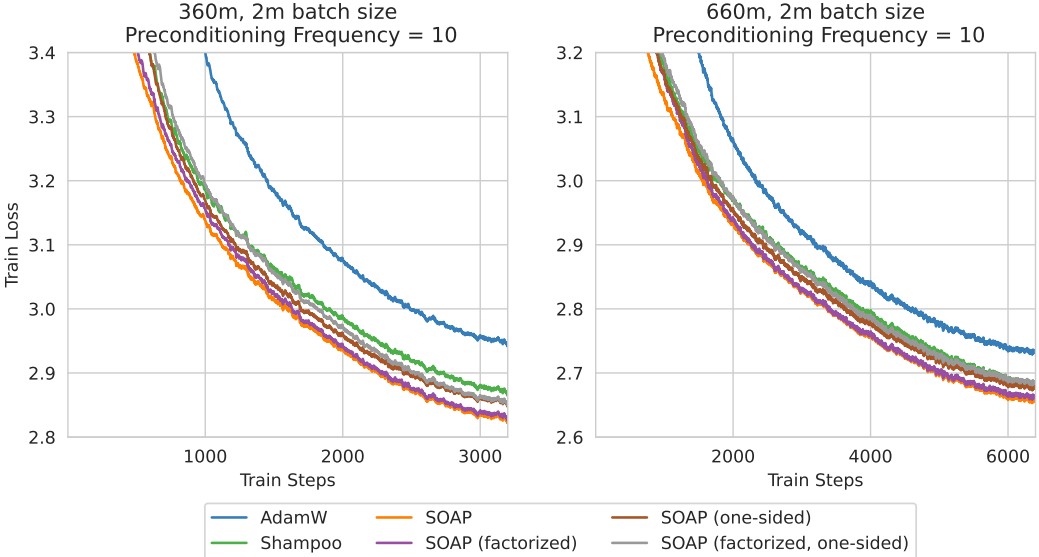

Figure 5: Performance of variants of SOAP which improve space usage or time overhead. 1. SOAP (factorized): Uses Adafactor instead of Adam in Shampoo's eigenbasis and 2. SOAP (one-sided): Uses $Q = I$ (i.e. no rotation) on the large side of weight matrix and 3. SOAP (factorized, one-sided): Combines both of these changes. We observe that while using Adafactor instead of Adam causes a negligible increase in loss, using the one-sided variant causes a larger increase. However, the one-sided variant also has much larger reduction in time and space overhead. For computational benefits of these variants see Appendices B.2 and B.3.

### B.2  SPACE USAGE OF SOAP

For a $m \times n$ matrix where $m > n$ we require

$$2m^2 \text{ (for } L, Q_L) + 2n^2 \text{ (for } R, Q_R) + 3mn \text{ (for gradient, } M, V)$$

space usage[4] (beyond weights and activations), specifically for $L, Q_L, R, Q_R$, momentum $(M)$, AdamW's second order estimate $(V)$, and the gradient. This is the same space usage as Distributed-Shampoo while AdamW uses $3mn$.

---

[4]One $mn$ is for storing the gradients, this can be avoided (as long as there is no gradient accumulation) by applying gradients along with backprop (Lv et al., 2024b) but this is not implemented by default in standard deep learning frameworks such as PyTorch. Hence we will include this term in all of our calculations.

### B.2.1 IMPROVING SPACE USAGE OF SOAP

The most direct way to reduce memory is using low precision to store the $L, R, Q_L, Q_R, V$ matrices, which is done by Dettmers et al. (2022); Wang et al. (2024). Orthogonal to the low precision approaches, there are two algorithmic approaches to improving the space usage of SOAP:

- Using Adafactor instead of Adam as the diagonal preconditioner after rotating by $Q_L$ and $Q_R$. This reduces the space usage by $mn$.
- Using one sided version of SOAP (Appendix B.1). This reduces space usage from $2m^2 + 2n^2 + 3mn$ to $2\min(m,n)^2 + 3mn$.
- Combining these approaches yields space usage of $2\min(m,n)^2 + 2mn$.

For standard transformer architectures the last variant which combines the two approaches would yield less space usage overall compared to AdamW (which uses $3mn$).

We try these approaches in Figure 5. We observe that using Adafactor instead of AdamW yields very small reductions in performance while using one-sided preconditioner results in larger reductions. Nonetheless even after combining these two approaches the resulting optimizer outperforms AdamW while having a smaller space requirement than AdamW. Regarding space usage we also note that Adafactor (with momentum added back) itself utilizes only $2mn$ space usage and has been shown to perform comparable to AdamW for ViT training (Zhai et al., 2022) and for language model training (Zhao et al., 2024c). Further space reduction beyond Adafactor has been studied in the Adalomo (Lv et al., 2024a), GaLore (Zhao et al., 2024a), and AdaMeM (Vyas et al., 2024) papers.

### B.3 TIME OVERHEAD OF SOAP

There are two types of overhead of Shampoo and SOAP over AdamW: the overhead per step and the overhead when changing the preconditioner (or for SOAP, the preconditioner's eigenbasis). Let us first analyze the first one. For SOAP per step for a layer of size $m \times n$ we have an overhead of

$$m^3 \text{ (updating } L) + n^3 \text{ (updating } R) + (2m^2n + 2mn^2) \text{ (projecting and projecting back on both sides).}$$

We note that this is more than the overhead of Shampoo which is $m^3 + n^3 + m^2n + n^2m$. This can be observed in Figure 2 (bottom, right) but not in the other figures since there the second type of overhead is the dominant term.

The second type of overhead is due to changing the preconditioner for Shampoo (or for SOAP, preconditioner's eigenbasis i.e. $Q_L$ and $Q_R$). The DistributedShampoo (Shi et al., 2023) implementation of Shampoo uses a direct call to `torch.linalg.eigh` for this. Following Wang et al. (2024) we use Algorithm 4 which uses power iteration based approach which calls `torch.linalg.qr`. We note that `torch.linalg.qr` is faster than `torch.linalg.eigh` (Documentation, 2024). In Figure 6 (right) we see that using power iteration based approach (`torch.linalg.qr`) performs as well as fresh eigenvector decomposition (`torch.linalg.eigh`).

**Effect of frequency on overhead:** In Figure 6 (left), we observe that the overhead decreases as the preconditioning frequency increases, i.e., the frequency of invoking Algorithm 4. If the only additional computation occurred in Algorithm 4, we would expect the overhead to scale as $1.0/(\text{preconditioning frequency})$, approaching zero. However, empirical results (Figure 6 left) show that the overhead approaches an asymptote greater than zero. This is attributable to the additional matrix multiplications required to update $L$, update $R$, project the gradient, and reproject the gradient (for each layer) in the optimizer. Currently, these operations are performed in float32; reducing the precision of these operations, as proposed in Wang et al. (2024), could lower this asymptote.

### B.3.1 IMPROVING TIME OVERHEAD OF SOAP

The per step overhead of SOAP can be reduced by using low precision to store the $L, R, Q_L, Q_R, V$ matrices (Dettmers et al., 2022; Wang et al., 2024), which in turn will speed up computation done using these matrices. This approach cannot be used for reducing the overhead for the preconditioner update in popular deep learning frameworks such as Pytorch since `torch.linalg.qr` does not

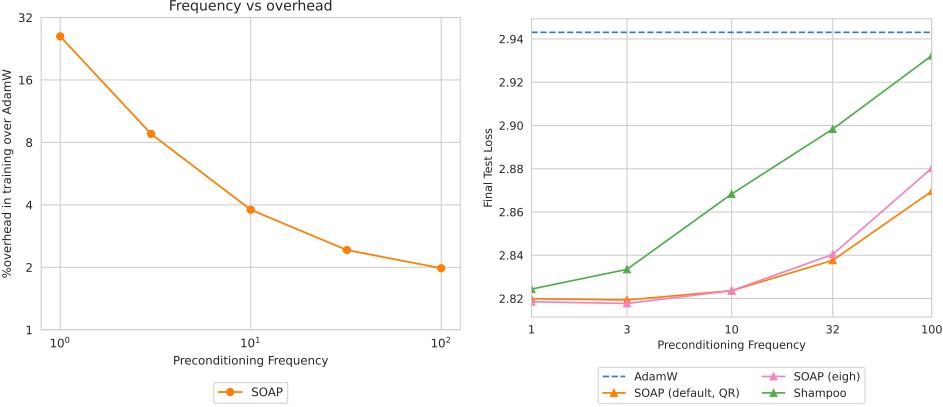

Figure 6: (Left) Depicting the overhead of SOAP over AdamW as a function of precondition-ing frequency (Right) Comparing the performance of SOAP with `torch.linalg.eigh` for computing the eigenvectors with Algorithm 4, which uses `torch.linalg.qr`. Note that `torch.linalg.qr` is computationally more efficient than `torch.linalg.eigh` (as men-tioned in Documentation (2024)); however, both seem to have comparable performance throughout the preconditioning frequency spectrum.

support precision lower than `float32`. Orthogonal to the low precision approach we can improve the per step time overhead of SOAP by the following algorithmic approaches:

- Using Adafactor instead of Adam (Appendix B.2) as the diagonal preconditioner after ro-tating by $Q_L$ and $Q_R$. In this version of SOAP the overhead can be improved by from $m^3 + n^3 + 2m^2n + 2n^2m$ to $m^3 + n^3 + m^2n + n^2m + \max(m, n)^2 \min(m, n) + \min(m, n)^3$ by merging the project and project back steps for the smaller dimension.

- Using one sided version of SOAP (Appendix B.1). This reduces overhead from $m^3 + n^3 + 2m^2n + 2n^2m$ to $\min(m, n)^3 + 2\min(m, n)^2 \max(m, n)$.

- Combining these approaches yields an overhead of $\min(m, n)^2 \max(m, n) + 2\min(m, n)^3$

Using one-sided version also reduces the second type of overhead from a calls to `torch.linalg.qr` on a $m \times m$ and a $n \times n$ matrix to only a single call to $\min(m, n) \times \min(m, n)$ matrix.

## C  LONGER DURATION RUN

Chinchilla scaling laws (Hoffmann et al., 2022) suggest that it is compute optimal to use tokens which are approximately 20x the models size, which is what we have been using for our standard runs. But many recent LLMs such as the LLaMA (Touvron et al., 2023) series of models are trained on much larger token counts. This can be to take into account the computational cost during infer-ence (Sardana et al., 2024) or to create models which are usable or finetunable by downstream users. In Figure 7 we train a language model with AdamW on a 100x model size token count. We then train the same model with SOAP for 50x, 75x, and 100x token counts to approximate the efficiency benefits. We find efficiency benefits ($> 40\%$) similar to those observed in Figure 2 for AdamW runs with 20x token counts.

## D  GALORE

We tried GaLore for 210m model, and while it outperformed AdamW it performed worse than Shampoo. Hence we do not try GaLore for higher model sizes.

**Hyperparameter sweeps:** We did the following sweeps:

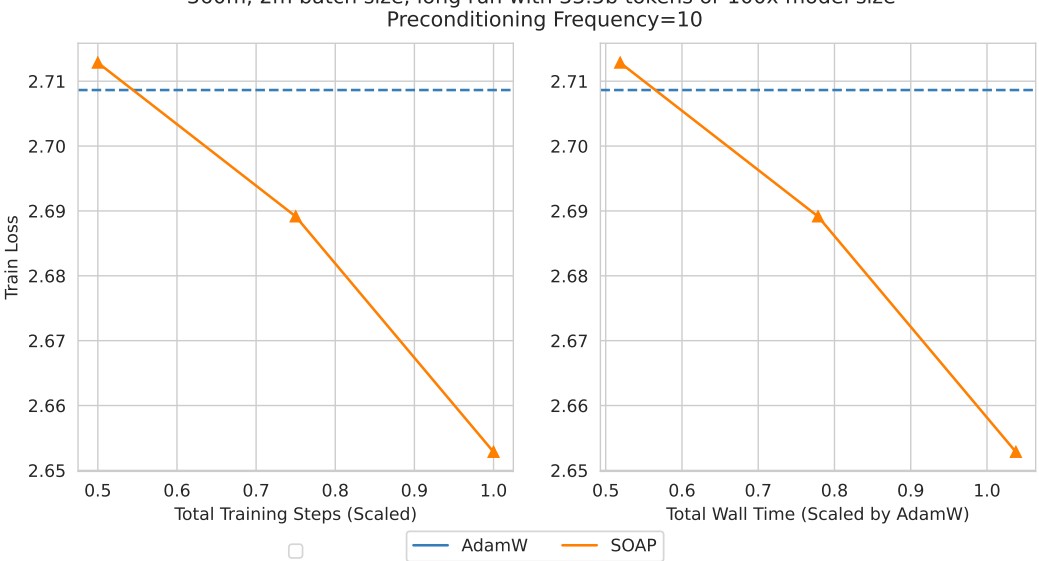

Figure 7: Total training steps (Left) and total wall clock time versus final test loss for long runs (#tokens = 5x "chinchilla" tokens = 100x model size).

1. We swept the cross product over learning rate $(3.16e-4, 1e-3, 3.16e-3, 1e-2)$, preconditioning frequency $(10, 50, 200)$, both sided and one sided versions. Frequency 200 had the best results matching the observation of Zhao et al. (2024a).

2. We did a cross product sweep over learning rate $(3.16e-4, 1e-3, 3.16e-3, 1e-2)$, both sided and one sided versions with $\beta_2 = .99$ instead of $.95$ and preconditioning frequency 200.

3. We did a cross product sweep over learning rate $(3.16e-4, 1e-3, 3.16e-3, 1e-2)$, both sided and one sided versions, preconditioning frequency $(50, 200)$ with $\beta_1 = .9$ instead of $.95$.

The best performing run among all of these achieved a final loss of 3.12 while the best Shampoo run achieved a final loss of 3.10.

