# OpenReview forum: "SOAP: Improving and Stabilizing Shampoo using Adam for Language Modeling"
_ICLR.cc/2025/Conference — ICLR 2025 Poster_

### Official Review · Reviewer_SFbv · 2024-10-23

**Soundness:** 4
**Presentation:** 4
**Contribution:** 4
**Rating:** 8
**Confidence:** 2

**Summary:**

I would like to preface my review: I am not an expert in this area. I have published a single paper in this area a very long time ago, but have not touched optimization since. I have asked several of my colleagues, who are among the most qualified researchers to review this paper, and they responded with very positive feedback. Still, I ask the AC to down-weight my review and defer to the other reviewers who may be more familiar with this space.

This work proposes a new optimization algorithm, called SOAP, that extends several of the (largely untested) concepts presented in Appendix B of the second Shampoo paper. In particular, they make changes to the computation of the Shampoo preconditioners---using dataset averages rather than running averages and computing the eigenvectors of L and R every $f$ steps---that make this algorithm now work.

Additionally, the authors prove a correspondence between the Shampoo parameter update and running Adafactor in the eigenbasis of Shampoo's preconditioner.  They then extend this to running Adam in Shampoo's preconditioner eigenbasis, and provide experimental results that demonstrate the effectiveness of their optimization algorithm.

**Strengths:**

This paper is exceptionally well written and easy to follow. Prior and related work are excellently summarized, and the contribution of work in the context of the overarching field is clear. The experimental results evaluate on a very relevant setting (LLM pre-training), the baselines are appropriate (direct comparison with Shampoo and AdamW), and the results demonstrate that SOAP is quite effective. I also appreciate the discussion of the hyperparameter sweeps in the appendix -- not conducing this comparison has caused some unfortunate situations in the past within the optimization community.

As an aside, the naming is excellent (shampoo, rinse, lather, and now soap...).

**Weaknesses:**

To be frank, I do not see any substantial weaknesses. One could argue that it doesn't go far enough in certain directions, but these areas are acknowledged by the paper and "left to future work" (which I believe is appropriate given this paper's scope). This is, in my view, a very good paper.

**Questions:**

The authors write Soap is equivalent to running Adam in a rotated space; however, is transformation to the eigenbasis $G' \leftarrow Q_L^TGQ_R$ necessarily unitary? What is the specific operation in SO(d)?

Several nits:

Using $\phi$ rather than $\mathcal{L}$ or $L$ to denote the loss seems unusual (although perhaps this is to distinguish from the Shampoo preconditioner $L$).

---

> ### Author Response · Authors · 2024-11-18
>
> Thanks for championing the paper. We are happy to see that the reviewer finds the work clear and easy to follow, and considers the work to have significant contribution to the deep learning optimization community. Specific comments are addressed below:
>
> > The authors write SOAP is equivalent to running Adam in a rotated space; however, is the transformation $G' \to Q_L^T G Q_R$ to the eigenbasis necessarily unitary? What is the specific operation in SO(d)?
>
> As pointed out, we project the gradient as $G' \leftarrow Q_L^T G Q_R$. If we represent $g = vec(G)$ and $g' = vec(G')$, where $vec(A)$ represents vectorization of the matrix $A$, then the update can be written as $g' \leftarrow (Q_L \otimes Q_R) g$, where $\otimes$ represents the Kronecker product. $Q_L \otimes Q_R$ represents the orthogonal transformation.
>
> We hope that we have addressed your questions and concerns. If not, we would be happy to answer any additional questions.

---

> > ### Comment · Reviewer_SFbv · 2024-11-27
> >
> > Thanks for follow-up! I've read the other reviews (and rebuttals) for this submission and maintain my score.

---

### Official Review · Reviewer_vXc9 · 2024-10-29

**Soundness:** 3
**Presentation:** 3
**Contribution:** 2
**Rating:** 6
**Confidence:** 5

**Summary:**

In this paper, the authors propose SOAP, ShampoO with Adam in the Preconditioner’s eigenbasis. They starts with an insight: Shampoo is equivalent to running Adafactor in the eigenbasis of Shampoo’s preconditioner. SOAP continually updates the running average of the second moment as Adam does.In the large batch regime, SOAP reduces the number of iterations by over 40% and wall clock time by over 35% compared to AdamW, with approximately 20% improvements in both metrics compared to Shampoo.

**Strengths:**

* Interesting connection and insights between Adafactor in Shampoo’s eigenspace and SOAP
* The name is funny
* Experiment results seem pretty promising

**Weaknesses:**

* I am a bit confused by the main contribution of this paper overall, algorithmetic insights on connection between adafactor and shampoo are interesting, hence I am recommending acceptance. However, beyond that, the rest seems to be a special case of galore [see my comments in the question below.]. Maybe a more efficient implementation?
* As far as I know, in pytorch, SVD was also implemented via power iteration but at cuda kernel level. The difference proposed here seems to be how many iterations it runs.
* Please also do some downstream evaluations on checkpoints output by different optimizers, just for the sake of completeness.

**Questions:**

* In theory, how is this different from running GaLore under "full" mode + adam/adafactor? Please refer to the following code snippet  in official galore repo for the context:
```python
low_rank_grad = torch.matmul(self.ortho_matrix[0].t().to(full_rank_grad.device.type), full_rank_grad) @ self.ortho_matrix[1].t().to(full_rank_grad.device.type)
```
https://github.com/jiaweizzhao/GaLore/blob/2cc66f88cce189e505affbb91042a8e77f5bf4e9/galore_torch/galore_projector.py#L39

* What happens if you use power iterations to approximate SVD in galore, does that still work? it feels like that should be included as a baseline too. [1] seems to suggest it works?

* Why does higher frequency lead to worse results? Intuitively, it should provide better and more up-to-date approximation to the eigenspace?

[1] Liang, Kaizhao, et al. "Memory-efficient llm training with online subspace descent." arXiv preprint arXiv:2408.12857 (2024).

---

> ### Author Response · Authors · 2024-11-18
>
> We thank the reviewer for their valuable and insightful feedback. We are happy to see that the reviewer finds the connection between Shampoo and Adafactor novel, and our experimental results promising. Specific comments are addressed below:
>
> > I am a bit confused by the main contribution of this paper overall; algorithmic insights on the connection between Adafactor and Shampoo are interesting. However, beyond that, the rest seems to be a special case of GaLore.
>
> As detailed in Section 3 (Lines 169-179), SOAP differs from GaLore in two key aspects:
>
> 1. The difference in how projection matrices are computed - GaLore uses SVD of the current gradient, while we maintain an exponential moving average of $GG^T$ and $G^T G$.
> 2. Projecting the momentum on the preconditioned space, every time the preconditoning space is updated. GaLore does not do this.
>
> Appendix D shows these technical differences contribute to the superior performance of SOAP as compared to GaLore.
>
> > What happens if you use power iterations to approximate SVD in GaLore; does that still work? It feels like that should be included as a baseline too. [1] Liang, Kaizhao, et al. "Memory-efficient LLM training with online subspace descent." arXiv preprint arXiv:2408.12857 (2024).
>
> Including GaLore with power iteration approximations as a baseline is an interesting idea. But we note that even without any approximations GaLore performs worse than Shampoo in our experiments (Appendix D).
>
> > Why does higher frequency lead to worse results? Intuitively, it should provide a better and more up-to-date approximation to the eigenspace.
>
> We apologize for the confusion. The term “preconditioning frequency” refers to the delay between updates. For a frequency of $k$, the preconditioner is updated once every $k$ steps. Higher values of $k$ reduce computational cost but lead to less frequent updates, which can degrade performance.
>
> > Please also do some downstream evaluations on checkpoints output by different optimizers, just for the sake of completeness.
>
> Yes, we will add these evaluations in future revisions.
>
> We hope that we have addressed your questions and concerns. If not, we would be happy to answer any additional questions.

---

> ### Author Response · Authors · 2024-11-23
>
> We would like to thank the reviewer again for their valuable feedback. We would be grateful if the reviewer would consider updating their score based on our response. If not, we would be happy to answer any additional questions.

---

> ### Comment · Reviewer_vXc9 · 2024-11-23
> **Re: "GaLore performs worse than Shampoo"**
>
> "Including GaLore with power iteration approximations as a baseline is an interesting idea. But we note that even without any approximations GaLore performs worse than Shampoo in our experiments (Appendix D)."
>
>
> I don't think this is a valid argument. Since single step power iteration is not as expensive, you can afford to run it every step to get more up-to-date information of the projected space. Whether it's better or worse is uncertain, but it should be a very obvious baseline.

---

> ### Author Response · Authors · 2024-11-23
>
> We tried preconditioning frequency of 10, 50 and 200 for GaLore (as mentioned in Appendix D) and found* that 200 preconditioning frequency performs the best. This shows that GaLore does not benefit from updating the preconditioning matrices faster. By "performs the best" we refer to final loss given a fixed number of steps (not wall clock time). Quoting appendix D for completeness:
>
> "We swept the cross product over learning rate (3.16e − 4, 1e − 3, 3.16e − 3, 1e − 2), preconditioning frequency (10, 50, 200), both sided and one sided versions. Frequency 200 had the best results matching the observation of Zhao et al. (2024a)."
>
> We hope this answers your question, please let us know if you have any further questions.
>
> *This is likely because GaLore maintains momentum in the rotated (by eigenvectors) space and hence at lower frequencies the momentum estimates possibly become quite bad whenever the eigenvectors change.

---

> > ### Comment · Reviewer_vXc9 · 2024-11-25
> >
> > That is an interesting observation. So SOAP doesn't suffer from the frequent update problem, but Galore does? Why is that the case?

---

> ### Author Response · Authors · 2024-11-25
>
> We think this likely because GaLore maintains momentum in the rotated (by eigenvectors) space (while SOAP maintains it in the original space) and hence at lower frequencies the momentum estimates possibly become quite bad for GaLore whenever the eigenvectors change.

---

> > ### Author Response · Authors · 2024-12-01
> >
> > As the discussion period ends tomorrow, we would like to thank the reviewer for their feedback and kindly ask them to let us know if our responses have addressed their questions and concerns. If so, we would be grateful if the reviewer could update their score accordingly.

---

> > > ### Comment · Reviewer_vXc9 · 2024-12-02
> > >
> > > Thanks for the explanations. Though I still have some doubts to some unfalsifiable claims, I would still recommend this paper for acceptance by keep my score due to its interesting theoretical connections. Best of luck!

---

> ### Author Response · Authors · 2024-12-02
>
> Could you please let us know what the unfalsifiable claims are?
>
> We note that the optimizer code is available at https://anonymous.4open.science/r/SOAP-F93B/soap.py.

---

> > ### Comment · Reviewer_vXc9 · 2024-12-02
> >
> > "We think this likely because GaLore maintains momentum in the rotated (by eigenvectors) space (while SOAP maintains it in the original space) and hence at lower frequencies the momentum estimates possibly become quite bad for GaLore whenever the eigenvectors change."
> >
> > I can see this as a plausible explanation. But it's unfalsifiable in the sense that we don't enough time to really run ablation studies to test this theory.

---

> > > ### Author Response · Authors · 2024-12-02
> > >
> > > We have clearly stated the differences between our implementation and GaLore in section 3 and these are the only potential reasons for why GaLore does not work as well as SOAP. We also note that why GaLore does not perform well is not the focus of our work. Furthermore, we would have been willing to run more experiments had the reviewer mentioned this concern earlier. In any case we will add more ablations with GaLore in the final version.
> > >
> > > In case the reviewer's concern is that power iteration can outperform exact eigenvector decomposition we would like to mention that Figure 6 shows that whether we run power iteration or exact eigenvector decomposition does not affect SOAP's performance at frequencies <= 32.

---

### Official Review · Reviewer_C64v · 2024-11-01

**Soundness:** 3
**Presentation:** 3
**Contribution:** 3
**Rating:** 6
**Confidence:** 3

**Summary:**

The paper establishes a connection between Shampoo and Adafactor and shows that Shampoo with the ½ power is equivalent to running Adafactor in the eigenbasis of Shampoo’s preconditioner.
Building on this insight the paper proposes to continually update the running average of the second moment, just as Adam, but in the slowly changing coordinate basis.
The new optimizer SOAP only introduces one additional hyperparameter (the preconditioning frequency) compared to Adam.
In language modeling experiments up to model sizes of 660m parameters the authors demonstrate that SOAP outperforms Adam and Shampoo on pre-training tasks.

**Strengths:**

This paper draw a novel connection between Shampoo and Adafactor, which has not been done before (to the best of my knowledge).
The paper is very well written and clear in the description of its method. The authors also provide code.
In general the conducted experiments are very thorough (especially the experiments on batch size scaling and efficiency) and demonstrate the benefits of SOAP compared to Adam or Shampoo. They authors also provide detailed analysis of the memory consumption of SOAP in the Appendix.

**Weaknesses:**

The main weakness of this paper (which is also acknowledged by the authors) are the rather small scale experiments in language model pre-training compared to current state of the art or production models.
I believe that showing good performance compared to AdamW on the 1B to 7B parameter scale, would strongly support the adoption of SOAP.
Also, it would be interesting how the new optimizer SOAP performs in combination with newly emerging Linear Recurrent Models (like Mamba).

**Questions:**

- L.350: n -> In (typo)
- Did you observe any influence of SOAP on training instabilities (e.g. grad norm slikes / rise or loss spikes) in language model pretraining?
- It would be helpful for others adopting SOAP if the authors provide learning curves for other common metrics monitored during pre-training (e.g. parameter norms, gradienten norms, validation loss, etc.)? This would help compering other training setups to your suggested one.

---

> ### Author Response · Authors · 2024-11-18
>
> We thank the reviewer for their valuable and insightful feedback. We are happy to see that the reviewer finds the paper well-written and novel. Their specific comments are addressed below:
>
> > I believe that showing good performance compared to AdamW on the 1B to 7B parameter scale, would strongly support the adoption of SOAP. Also, it would be interesting how the new optimizer SOAP performs in combination with newly emerging Linear Recurrent Models (like Mamba).
>
>  We agree that validating SOAP on larger model scales and alternative architectures is important. Due to computational constraints, we focused on 360M and 660M models in this study, but we plan to extend our work to larger scales and diverse model architectures in the future.
>
> > Did you observe any influence of SOAP on training instabilities (e.g., grad norm spikes / rise or loss spikes) in language model pretraining?
>
> No, we did not monitor grad norm spikes during training.
>
>
> > It would be helpful for others adopting SOAP if the authors provide learning curves for other common metrics monitored during pre-training (e.g., parameter norms, gradient norms, validation loss, etc.). This would help comparing other training setups to your suggested one.
>
> We are happy to include these learning curves in a future revision to support adoption and facilitate comparisons.
>
> We hope that we have addressed your questions and concerns. If not, we would be happy to answer any additional questions.

---

> ### Author Response · Authors · 2024-11-23
>
> We would like to thank the reviewer again for their valuable feedback. We would be grateful if the reviewer would consider updating their score based on our response. If not, we would be happy to answer any additional questions.

---

### Official Review · Reviewer_kaMq · 2024-11-04

**Soundness:** 2
**Presentation:** 2
**Contribution:** 2
**Rating:** 5
**Confidence:** 3

**Summary:**

This paper introduces SOAP (ShampoO with Adam in the Preconditioner’s eigenbasis), an optimizer designed to improve the efficiency and stability of Shampoo by integrating Adam updates within Shampoo's eigenbasis. SOAP simplifies Shampoo by reducing its hyperparameters and computational overhead while maintaining its benefits.

**Strengths:**

(1)  This paper is well-organized.

(2) This work has significant implications for large language model training.

**Weaknesses:**

(1) The evaluation in this paper is limited, as it only tests the proposed SOAP on 360M and 660M language models. It is unclear whether SOAP can scale effectively to larger models, such as 1B or 7B, or how it performs in fine-tuning settings.

(2) Given that SOAP is intended as a general optimizer, it would be valuable to assess its performance across diverse tasks, including vision and graph tasks, to better understand its generalization.

(3) While the paper establishes a theoretical connection between Shampoo and Adafactor that motivates SOAP, it remains unclear why SOAP outperforms both Shampoo and Adam.

(4) Since SOAP is inspired by the connection between Adafactor and Shampoo, it is essential to include Adafactor as a baseline to fully evaluate SOAP's effectiveness.

**Questions:**

(1) What's the performance of SOAP in other tasks such as vision and graph tasks?

(2) why can SOAP perform better than Shampoo? Is there any rationale behind this?

(3) What's the performance of SOAP in fine-tuning settings?

---

> ### Author Response · Authors · 2024-11-18
>
> We thank the reviewer for their valuable and insightful feedback. We are happy that the reviewer finds the paper well-organized and of significant importance for the LLM community. The specific comments are addressed below:
>
> > The evaluation in this paper is limited, as it only tests the proposed SOAP on 360M and 660M language models. It is unclear whether SOAP can scale effectively to larger models, such as 1B or 7B, or how it performs in fine-tuning settings.
>
>  We acknowledge that our experiments are limited to 660M-scale models due to computational constraints, as mentioned in the paper (Line 530). However, given the theoretical foundations of SOAP and its promising results at this scale, we are optimistic about its scalability to larger models. Validating SOAP at larger scales remains a priority for future work. We also hope to conduct fine-tuning experiments in the future.
>
> > Given that SOAP is intended as a general optimizer, it would be valuable to assess its performance across diverse tasks, including vision and graph tasks, to better understand its generalization.
>
> Currently, our focus is on language modeling, which we believe is an important task in itself. To align with this focus, we can revise our contributions to explicitly state applicability to language modeling setups. We plan to explore SOAP’s performance in other domains as part of future work.
>
> > While the paper establishes a theoretical connection between Shampoo and Adafactor that motivates SOAP, it remains unclear why SOAP outperforms both Shampoo and Adam.
>
> As shown in Figure 5 in the Appendix (Line 936), factorized SOAP (i.e., Adafactor in Shampoo’s eigenbasis) performs comparably to SOAP. However, Shampoo only matches the performance of factorized SOAP when using a preconditioning frequency of 1, which incurs significant computational overhead. The main contribution of this work is to decouple the eigenbasis updates from eigenvalue updates, enabling SOAP (or factorized SOAP) to achieve comparable performance even at higher preconditioning frequencies (due to adaptivity in the eigenvalues).
>
> > Since SOAP is inspired by the connection between Adafactor and Shampoo, it is essential to include Adafactor as a baseline to fully evaluate SOAP's effectiveness.
>
> As shown in recent work [1], the performance of Adafactor and Adam is comparable to each other. We would be happy to include Adafactor as a baseline if this is critical for the reviewer's evaluation of the work.
>
> [1] - Zhao et al. 2024 - Deconstructing What Makes a Good Optimizer for Language Models
>
>
> > What's the performance of SOAP in other tasks such as vision and graph tasks, and fine-tuning settings?
>
> Evaluating SOAP in these tasks is part of our planned future work.
>
> > why can SOAP perform better than Shampoo? Is there any rationale behind this?
>
> SOAP performs better than Shampoo at preconditioning frequency greater than 1. This is because within preconditioning steps, even though the eigenbasis remains fixed, SOAP still updates the eigenvalues, leading to more adaptivity as compared to Shampoo.
>
> > What's the performance of SOAP in fine-tuning settings?
>
> Evaluating SOAP for fine-tuning settings is a part of future work.
>
> We hope that we have addressed your questions and concerns. In light of this, we would be grateful if you would consider updating your review of the paper. Alternatively, we would be happy to answer any additional questions that lead you to maintain your current recommendation.

---

> > ### Author Response · Authors · 2024-11-23
> >
> > We would like to thank the reviewer again for their valuable feedback. We would be grateful if the reviewer would consider updating their score based on our response. If not, we would be happy to answer any additional questions.

---

> > > ### Comment · Reviewer_kaMq · 2024-11-25
> > >
> > > Thank you for the author's response. However, I respectfully disagree with the claim that this work focuses on LLMs, as I did not observe any modifications to SOAP that are specifically tailored for LLMs. While the theoretical analysis is noteworthy, I believe the experimental validation is equally important. It remains unclear whether SOAP can perform effectively on large-scale LLMs, e.g. with 1B parameters, and on other tasks such as classification on ImageNet. As such, I will raise the rating to 5 but I am unable to revise my rating to acceptance at this time.

---

### Public Comment · ~Kyunghun_Nam1 · 2025-06-06
**Some questions related to Algorithm2**

Dear, I hope this message finds you well.

I read your paper with great interest, and I have a question about something in Algorithm 2 that is disturbing my understanding, albeit in a minor way.

The key is that in lines 10, 11, and 12, the matrices are not matched in size, so operations like matrix multiplication are not well defined.

Line 10: Trying to multiply a $m \times n$ matrix by a $m \times 1$ column vector.
Line 11: Attempting to multiply a $1 \times n$ row vector by an $m \times n$ matrix.

Even if you consider these two typos, $AC^T$ is not defined. Since $A$ is an $m \times 1$ column vector and $C$ is a $1 \times n$ row vector, $AC$ seems to be the correct notation.

Also, in Line 14 (Projecting back to original space), why $Q_L^T G'' Q_R$? I think $Q_L G'' Q_R^t$ is correct, isn't it?

I would appreciate an answer. Thank you.

---

> ### Public Comment · ~Depen_Morwani1 · 2025-06-07
>
> Hi Kyunghun Nam
>
> Thanks a lot for looking at the paper. Yes, these are indeed typos in the Algorithm.
>
> Line 10: It should be $n \times 1$ column vector.
>
> Line 11: It should be $1 \times m$ row vector.
>
> Line 12: It should be AC.
>
> Line 14: It should be $Q_L G_t'' Q_R^T$.
>
> Thanks a lot for catching these typos. We will fix them in the next revision.

---

### Meta-Review · Area_Chair_RpeZ · 2024-12-19

**Metareview:**

This paper proposes SOAP, a novel optimization method that builds on the insight that Shampoo is equivalent to running Adafactor in the eigenbasis of Shampoo’s preconditioner. SOAP updates the running average of the second moment in a manner similar to Adam, achieving significant efficiency improvements in large-batch training.

The writing is clear, the method is well-described, and the authors provide thorough experiments, particularly on batch size scaling and efficiency, which highlight the benefits of SOAP.

However, the paper’s experimental setup has notable limitations. The scale of the experiments is relatively small, the settings are not practical for real-world applications, and there is a lack of downstream task evaluations or fine-tuning results. More practical results are particularly concerning, as lower loss does not always correlate with better performance, especially in fine-tuning scenarios. Moreover, the experiments are verified on the limited types of tasks, limiting the generalizability of the results.

Despite these weaknesses, the paper introduces a highly novel idea that bridges the gap between first-order and second-order optimizers effectively. Given its theoretical contribution and potential practical impact, I recommend weak acceptance.

**Additional Comments On Reviewer Discussion:**

During the rebuttal period, most reviewers maintained a positive stance on the paper. Several reviewers requested additional experimental results, particularly on downstream tasks, to better evaluate the method’s practical impact. However, the authors chose to defer these results to future work. This decision did not significantly affect the reviewers’ opinions, which remained generally favorable. Overall, the discussion during the rebuttal reinforced the initial inclination to recommend acceptance.

---

### Decision · Program_Chairs · 2025-01-22

Accept (Poster)